# Nutrient Intake with Early Progressive Enteral Feeding and Growth of Very Low-Birth-Weight Newborns

**DOI:** 10.3390/nu14061181

**Published:** 2022-03-11

**Authors:** Rasa Brinkis, Kerstin Albertsson-Wikland, Rasa Tamelienė, Asta Vinskaitė, Kastytis Šmigelskas, Rasa Verkauskienė

**Affiliations:** 1Department of Neonatology, Lithuanian University of Health Sciences, 44307 Kaunas, Lithuania; rasa.tameliene@lsmuni.lt (R.T.); vinskaite@gmail.com (A.V.); 2Department of Physiology/Endocrinology, Institute of Neuroscience and Physiology, Sahlgrenska Academy, University of Gothenburg, 40530 Gothenburg, Sweden; kerstin.albertsson.wikland@gu.se; 3Health Research Institute, Faculty of Public Health, Lithuanian University of Health Sciences, 44307 Kaunas, Lithuania; kastytis.smigelskas@lsmuni.lt; 4Institute of Endocrinology, Lithuanian University of Health Sciences, 44307 Kaunas, Lithuania; rasa.verkauskiene@lsmuni.lt

**Keywords:** very low birth weight, newborn, growth, enteral feeding, nutritional intake, extrauterine growth restriction

## Abstract

Early nutrition is one of the most modifiable factors influencing postnatal growth. Optimal nutrient intakes for very preterm infants remain unknown, and poor postnatal growth is common in this population. The aim of this study was to assess nutrient intake during the first 4 weeks of life with early progressive enteral feeding and its impact on the in-hospital growth of very low-birth-weight (VLBW) infants. In total, 120 infants with birth weights below 1500 g and gestational ages below 35 weeks were included in the study. Nutrient intakes were assessed daily for the first 28 days. Growth was measured weekly until discharge. Median time of parenteral nutrition support was 6 days. Target enteral nutrient and energy intake were reached at day 10 of life, and remained stable until day 28, with slowly declining protein intake. Median z-scores at discharge were −0.73, −0.49, and −0.31 for weight, length, and head circumference, respectively. Extrauterine growth restriction was observed in 30.3% of the whole cohort. Protein, carbohydrates, and energy intakes correlated positively with weight gain and head circumference growth. Early progressive enteral feeding with human milk is well tolerated in VLBW infants. Target enteral nutrient intake may be reached early and improve in-hospital growth.

## 1. Introduction

Premature birth results in a sudden abruption of nutrients delivered via the placenta. Many very low-birth-weight infants (VLBW, <1500 g) are born at a time of rapid fetal growth and high nutritional demand. The goal to provide adequate amounts of nutrients in the first days of life for these patients is challenging, yet of high importance. Extrauterine growth restriction (EUGR, z-scores: <−1.28 at discharge) is common in this population. Ideally, the postnatal growth of a premature infant should be equivalent to the intrauterine growth of the normal human fetus [1]. However, the environment for extrauterine growth is much different from the intrauterine, and the majority of very preterm infants do not reach their birth centile at term-corrected gestational age. Cardiorespiratory instability, stress, pain, comorbidities, and genetics impact postnatal growth [2,3,4], and an optimal growth pattern promoting long-term health and development has not yet been defined.

Early nutrition is one of the most modifiable factors influencing postnatal growth. Despite many nutritional studies performed in recent years, the optimal nutrient intake for very preterm infants remains unknown. The high variability of nutritional calculations and growth assessment makes the meta-analysis of nutritional studies unreliable [5]. Clinicians may control nutrient intake with parenteral nutrition and enteral feedings, but it is not clear what amount of each component is ideal to promote desired growth velocity, body composition, and long-term developmental outcomes. Nutritional policies vary among institutions [6]. Own mothers’ milk (OMM) is the best food for all infants, but enteral nutrition is often delayed in very premature newborns. To prevent nutritional deficits, parenteral nutrition (PN) is essential for critical periods of the premature infant’s life, such as the first days after birth or during illness. However, PN is related to certain complications and adverse effects, such as catheter-related infections, intravenous glucose and lipid intolerance, and liver toxicity. Thus, enteral nutrition (EN) should be started as early as possible, preferably on the first day of life, and rapidly increased. There are well-defined benefits of own mother’s milk colostrum, and this important feeding step should not be missed [7,8]. 

Breast milk has multiple short- and long-term benefits on lower rates of late onset sepsis (LOS), necrotizing enterocolitis (NEC), retinopathy of prematurity (ROP), and bronchopulmonary dysplasia (BPD), as well as better cognitive outcomes [9]. Advancing enteral feeding early is challenging for clinicians because of uncertainty about feeding tolerance and concerns of potential NEC. However, studies did not show an increased risk of NEC with the faster advancement of feeding [10]. In addition to all OMM advantages, its nutritional composition is ideal for term infants. The desirable growth of very premature infants may not be achieved with milk only, and fortification is recommended for infants less than 1800 g [11]. Standard fortification is most widely used, although it may not meet high nutritional demands, especially in extremely low-birth-weight (ELBW) infants [12]. Individual fortification is recommended, but there is a lack of data on ideal timing, fortifiers, and milk volumes to start [13,14]. The aim of this study was to assess nutrient intake during the first 4 weeks of life with early progressive enteral feeding and its impact on the in-hospital growth of very low-birth-weight infants.

## 2. Materials and Methods

### 2.1. Study Population

A prospective, non-interventional observational study was conducted in the Hospital of Lithuanian University of Health Sciences Kauno klinikos, Department of Neonatology. Approval for the study was obtained at the Kaunas Regional Bioethics Committee (approval No. BE-2-12). The study was registered at ISRCTN Database (No. ISRCTN64647571). VLBW newborns (<1500 g) born between 31 May 2018 and 17 May 2020 were included into the study. The written consent of both parents was obtained.

Eligibility criteria for study participants: 

Inclusion criteria: birth weight < 1500 g and gestational age ≤ 34 weeks. Gestational age (full weeks) was estimated by last menstrual period or ultrasound.

Exclusion criteria: chromosomal abnormalities, genetic syndromes affecting growth, absent parental consent, surgical intervention with partial bowel removal. 

During the study period 182 VLBW newborns were born, and 120 VLBW newborns were included into the study. In total, 8 infants died before discharge, and 3 infants were excluded following partial bowel resection. In addition, 109 infants were discharged from the hospital, and 7 of them were discharged before 28 days after birth. The flowchart of the study subjects’ inclusion is shown in Figure 1.

### 2.2. Nutritional Practices

Parenteral nutrition. PN protocol was based on the European Society for Paediatric Gastroenterology Hepatology and Nutrition (ESPGHAN) recommendations [15]. PN was started right after birth as soon as peripheral or central vein was obtained. PN bags were made in Neonatal Intensive Care Unit using a closed system (Medimix Mini 4010, Impromediform, Lüdenscheid, Germany). An amino acid solution (Vaminolact 6.5%, Fresenius Kabi, Uppsala, Sweden), dextrose, and lipid emulsion (Smoflipid 20%, Fresenius Kabi, Uppsala, Sweden) were used as macronutrient sources. Pre-calculated standard PN solutions were prepared for most patients, with rare exceptions being clinical conditions requiring additional fluid restriction or PN intolerance confirmed by biochemical tests. A starter mixture containing 4.0 g/100 mL of amino acids and 10% dextrose was used for the first 24–48 h. On subsequent days, 3.3 g/100 mL of amino acids and 10–13.5% dextrose was used, and lower glucose concentration solution (6.5%) was used temporarily in the case of persistent hyperglycemia in extremely premature infants. On the first day of life, 2.5–3.0 g/kg of amino acids, 6.0–7.0 g/kg carbohydrates, 1.0 g/kg lipids, 45–50 kcal/kg, and 70–80 mL/kg of fluids were provided. Thereafter, full parenteral nutrition consisted of 3.5–3.8 g/kg/day of amino acids, 12.0–16.0 g/kg/day of carbohydrates, 3.0 g/kg/day of lipids, 80–90 kcal/kg, and 140 mL/kg of fluids.

Enteral nutrition. EN was targeted to ESPGHAN enteral nutrient supply recommendations [16]. EN was started in the first hours of life as soon as newborn’s condition was stabilized. Newborns with birth weight less than 1000 g were fed every 2 h until they reached 1000 g body weight, and newborns weighing more than 1000 g were fed every 3 h. Fresh OMM was a food of choice. If mother did not provide enough milk, donor human milk was administered. Donor human milk was processed at the Milk Bank, with donors being mothers who delivered prematurely. Maternal colostrum was applied orally before every feeding [17]. The starting volume was 20 mL/kg/day of milk, and feedings were advanced by 20–30 mL/kg/day as tolerated. Gastric residuals, regurgitating, and abdominal distension without tenderness were not considered as feeding intolerance. Vomiting, bloody stool or gastric residuals, abdominal distension with tenderness and cardiorespiratory instability, or any signs of suspected NEC were indications to withhold enteral feedings. Total enteral nutrition was achieved when the infant received 140–150 mL/kg/day of milk. Then, the volume was increased to 160–180 mL/kg/day. Formula was used only for a few days before discharge if the mother’s lactation could not be established to ensure full feedings.

Human milk fortifier was added to own mothers’ or donor milk when full enteral feeding was achieved. On the first 2 days of fortification, half-strength fortifier was added. If it was tolerated, then full-strength standard fortification was continued. Bovine-based powdered human milk fortifier (Aptamil FMS^®^, Milupa/Danone GmbH, Friedrichsdorf, Germany, Danone Nutricia, Cuijk, The Netherlands) was added to OMM or donor milk following the manufacturer’s guidelines (4.4 g of powder to 100 mL of milk, providing 1.1 g of additional protein, 2.7 g of carbohydrate, and 15 kilocalories).

### 2.3. Nutritional Calculations

Mother’s milk analysis. In the first days of life, all fresh unpasteurized OMM was used for enteral feedings to maximize benefits of colostrum. Mothers were encouraged to stay in the hospital 24/7, and they expressed milk every 3 h with the help of staff and lactation specialists. As soon as the mother was able to express extra milk for analysis (usually at day 5–7 after delivery), 1 mL of milk from each of eight fully expressed portions was taken, mixed well, and analyzed for average 24 h nutrient content. Milk analysis was carried out twice a week using a mid-infrared human milk analyzer (MIRIS, Uppsala, Sweden). All donor milk was also analyzed for macronutrients.

Nutritional intake was calculated from actual intake data obtained from medical records every 24 h, with a starting point of 12 PM. Total daily nutrient intake includes PN and EN intakes. For the first day’s parenteral intake, if the infant was born after 12 p.m. and did not receive PN for full 24 h, calculations were adjusted proportionally for the 24 h period. For parenteral protein, 1 g of parenteral amino acids equals 0.9 g of protein, as specified by manufacturer. For the first day’s enteral intake, the total amount of nutrients from milk received until 12 p.m. was calculated and not adjusted to the 24 h period. To calculate enteral nutritional intake on the days when OMM was analyzed, actual nutrient and calorie content was used. For the days between measurements, values were calculated with the exponential method between the two nearest values. For the first days when mother’s milk composition data were unavailable, OMM intakes were not included into total count of nutrients. Nutrient intake g/kg/day or kcal/kg/day were calculated for the birth weight initially, and for the actual body weight after birth weight was regained.

### 2.4. Anthropometric Measurements

Birth weight was recorded at the time of admission, and other anthropometric measurements were taken within 24 h after birth. Infants were weighed daily with incubator scales (Giraffe, GE Healthcare, Laurel, MD, USA) while in the incubator care, and portable electronic scales (Marsden, Rotherham, UK) when transferred to the cot. Scales were calibrated regularly according to the manufacturer’s instructions. Weight was measured to the nearest 5 g with incubator scales and 1 g with portable electronic scales. Body length and head circumference were measured weekly to the nearest 0.1 cm following the standard procedure [18]. An infant measuring rod (SECA, Hamburg, Germany) was used for length measurement, and non-stretchable single use tape (SECA, Hamburg, Germany) was used for head circumference measurement. Linear measurements were performed by two researchers to reduce errors. Length and head circumference were measured twice, and the average value was used for analyses. Z-scores were calculated using the Fenton calculator [19]. For infants discharged before 36 weeks of corrected gestational age, z-scores were calculated for the corrected corresponding week. For infants discharged at or after 36 weeks, z-scores were calculated for measurements at 36 weeks of corrected gestational age (CGA). Small for gestational age (SGA) at birth was defined by two methods—National reference [20] and Fenton reference [19]—when the infant’s weight was below the 10th percentile (National reference) or the z-score for weight was below −1.28 SDS (Fenton reference). Extrauterine growth restriction (EUGR) was diagnosed when the z-score for weight was below −1.28 SDS using the Fenton reference at 36 weeks of corrected gestational age or at discharge, whichever occurred earlier.

### 2.5. Data Analysis

Data analysis was performed using “Microsoft Excel” version 16.54 and “IBM SPSS Statistics” version 27. The normality of indicators was assessed using skewness and kurtosis and, in most cases, indicated the absence of Gaussian distribution. Therefore, the non-parametric approach in statistical analysis was used. The descriptive analysis included medians with interquartile ranges (IQR) for continuous variables and percentages for categorical indicators. For group comparisons, the non-parametric Kruskal-Wallis test was used. The associations between nutritional intake and anthropometric outcomes were assessed using the Spearman correlation.

## 3. Results

### 3.1. Sample Description

Since the sample was not homogenous for birth weight and gestational age, for more detailed analysis, we divided the whole cohort into three birth weight (BW) groups.

The main demographic characteristics of birth weight groups are presented in Table 1.

### 3.2. Nutritional Characteristics

Enteral feeding was started within 24 h after birth for all infants, except one who was in critical condition for the first 48 h. In total, 79.7% infants received maternal colostrum orally and enterally within 24 h after birth. The donor milk was used mainly for the first few days until mother’s lactation was established. In total, 17% of infants received donor milk beyond the first week. With the fast advancement of enteral feeds, the parenteral nutrition support was necessary within first week, with only 13.3% of infants requiring it longer, with maximum duration of 12 days. Median duration of parenteral nutrition support was 6 days. Standard human milk fortification was started when full enteral feeding was reached and continued until discharge. Nutritional characteristics of infants in different birth weight groups are presented in Table 2.

Among survivors without proven NEC diagnosis, only one infant required EN to be interrupted for more than 1 day before reaching full enteral feeds. After reaching full enteral feeds, two infants needed EN to be interrupted and parenteral nutrition re-established for more than 1 day. All three cases occurred because of suspected NEC, which was later not confirmed.

The NEC rate was 5.0%, and the SIP rate was 1.7%. Most cases (75%) occurred in infants of the lowest gestational age (22 to 26 gestational weeks) with many comorbidities and risk factors.

### 3.3. Availability of Mother’s Milk Data

To preserve the colostrum and own mother’s milk for early enteral feedings, mother’s milk samples were collected for analysis only when enough milk was left from each feeding portion. The number of samples collected each day post-partum are shown in Figure 2.

### 3.4. Daily Intake of Macronutrients and Energy

For the calculation of nutritional intake, only the known values of nutrients and calories were included. We did not include nutritional data from the mother’s milk intake if the milk composition was not known. For the first 3 days, parenteral nutrition was the main source of nutrients. The total nutrient and energy intake values from days 4 to 7, when the transition from PN to EN occurs and EN takes over PN fluids, are underestimated and represented mostly parenteral and donor milk intakes. The analysis of few milk samples does not reflect general data of early-stage milk composition and make nutritional calculations less accurate. Therefore, data of nutritional intake on days 4 to 7 are not presented.

Once full enteral feeds were established, the nutrient intake increased close to the recommended values, i.e., 3.5 g/kg/day of protein and 120 kcal/kg/day, both occurring on day 10 of life. Further intake of nutrients and calories remained stable until 28 days, except for a slowly declining protein intake. However, the recommended protein intake of 4–4.5 g/kg/day was reached only in 20% of ELBW infants using standard fortification. In total, 52.4% of infants (51% of ELBW) received less than 3.5 g/kg/day of protein on week 4 of life. The intake of carbohydrates exceeded recommended values. The daily intake of fluids and nutrients is shown in Figure 3.

### 3.5. Growth

Median z-scores in the whole cohort at discharge were −0.73, −0.49, and −0.31, and the changes in z-scores from birth to discharge were −0.66, −0.5, and −0.26 for weight, length, and head circumference, respectively. After initial postnatal weight loss during the first week of life, 87.5% of infants regained their birth weight within 2 weeks after birth. The weight z-score change from the second week onward was close to zero in all BW groups. Weight z-scores at discharge for groups <1000 g, 1000–1249 g, and 1250–1499 g were −1.18, −0.69, and −0.54, respectively. Length z-scores declined throughout hospitalization in all BW groups. A decline in head circumference (HC) z-scores was observed for the first 2 weeks in the smallest BW group but accelerated from day 14 and increased every week until discharge in all BW groups. HC growth of SGA infants accelerated markedly from second week after birth but slowed down between week 4 and discharge. 

Extrauterine growth restriction at 36 weeks corrected gestational age or at discharge was observed in 30.3% of the whole cohort and in 20.3% of non-SGA infants. For the growth patterns of infants in different BW groups, we separated 12 SGA infants (10% of the whole cohort according to the Fenton reference). The growth is shown in Figure 4, and the changes in z-scores from birth to 28 days and discharge are shown in Table 3.

### 3.6. Relationship between Nutrient Intake and Growth

We could not accurately estimate the correlation between nutrient intake and growth parameters during the first week of life since the true intake during this period was not known. During subsequent weeks, higher intakes of protein, carbohydrates, and calories were associated with better weight gain and head circumference growth in the whole cohort, especially during weeks 3 and 4, with the associations being weaker in newborns with BW < 1000 g. In SGA infants, higher nutrient intakes had a significant effect only on weight gain during weeks 3 and 4, and no effect on linear growth. Nutrient intake during the first 28 days did not differ between SGA and non-SGA infants. None of the nutrients had any significant correlation with the length growth during first 4 weeks after birth in the whole cohort. All correlations between nutrient intake and growth during the first 4 weeks in different BW groups and in the whole cohort are shown in the Figure 5. Weekly average nutrient intake in different birth weight groups and SGA infants is presented in Appendix A.

## 4. Discussion

### 4.1. Early Enteral Feeding

The results of this study show that the early start and rapid advancement of enteral feeding with human milk and fortification using standard protocol enables the achievement of the target enteral nutrient intake early and may improve the in-hospital growth of VLBW infants. Early progressive enteral feeds are well tolerated in VLBW and even in ELBW infants. A retrospective study conducted by Maas [21] and randomized study by Salas [22] reported similar results. There have been concerns about enteral feeding advancement and NEC, however, a Cochrane review did not find reduced NEC rates with slower advancement of enteral feeding [10]. The incidence of NEC across developed countries is 5–12% for VLBW infants and 7% globally [23,24]. In our study, the NEC rate in the whole cohort was at the lower end of those reported. 

A randomized study on parenteral nutrition delivery techniques previously conducted in our center reported a constant decline in the weight z-scores of VLBW infants during hospital stay. The weight z-scores in randomization groups were −1.44 and −1.12 at discharge, which are significantly lower compared to our cohort, However, this study did not analyze EN intake [25]. The duration of parenteral nutrition support was short and limited to the first week of life. Thus, we hypothesized that EN should have impacted growth more than PN. Nutritional practices in our institution were revised, and standard enteral feeding protocol was implemented. The aim of this study was to evaluate the nutritional intake of VLBW infants with early progressive enteral feeding and its role on in-hospital growth. 

Early progressive feeding strategies result in new challenges during the transition phase from parenteral to enteral nutrition. Studies have shown that the transition phase from parenteral to enteral nutrition results in nutritional deficits. However, these studies, parenteral support was longer, with the transition phase occurring in week 2 or 3 [26,27]. With the fast advancement of enteral feeding, this phase moves into the first week of life, when the relative restriction of fluids is recommended [15,28], and may result in even greater nutritional deficits. PN was dominant during the first 3 days in our study, and enteral fluid intake exceeded parenteral intake as early as day 4 after birth. We could not assess the actual nutrient intake during the first week, since we were able to analyze only a few specimens of colostrum which showed higher protein but lower fat and calorie content. It was shown that the use of more concentrated PN bags during the transition phase lessens nutritional deficits and improves growth [29]. These findings may be important in PN adjustments to ensure more nutrients with less fluids during the first week of life. On the other hand, all bioactive substances in fresh human milk and colostrum and its benefits to gastrointestinal adaptation, microbiota, and immunity may outweigh short-term nutritional deficits by contributing to long-term growth and development.

### 4.2. Nutritional Intake

In our cohort, the recommended enteral nutrient and energy intake was reached as early as the beginning of the second week (day 10 of life) and may have been reached even sooner with earlier fortification. EMBA recommends starting fortification at 50–80 mL/kg/day of milk [11]. However, these recommendations are based on studies with longer parenteral support and a later introduction of enteral feeds. In our study, 50–80 mL/kg/day of milk was reached at the day 3–4 of life. A current systematic review does not define the ideal time of human milk fortification [30,31]. Moreover, due to the lack of data on the safety of early introduction of bovine-based fortifiers and early exposure to cow milk protein in VLBW infants, in our institution, fortification is started when full enteral feeding is reached. Mother milk analysis showed a constant decrease in the protein content, reaching levels as low as 1.0–0.8 g/100 mL between days 28 and 42. It may suggest that the decline in all z-scores in SGA infants was caused by a cumulative protein deficit with standard fortification. However, the group of SGA infants in our cohort was too small (only 12 infants) to make reliable conclusions on nutritional and growth patterns. Randomized studies [32,33] have not shown better weight gain and head circumference growth with protein intake exceeding 4.0 g/kg/day. Targeting mainly protein intake values and not considering the weight and linear growth pattern may result in excessive unnecessary protein intake. Moreover, overfeeding in early infancy has been shown to be linked to later metabolic consequences [34,35].

Many nutritional studies to date have mainly focused on protein and energy intake. Most of calories come from carbohydrates and fat. Therefore, we looked at these nutrients’ intakes separately. Fat and energy intake remained stable until 28 days and met current recommendations. The intake of carbohydrates exceeded recommended values. Interestingly, the difference between the recommended and estimated intake was close to the amount of carbohydrate intake from human milk fortifier. Thus, targeting higher protein intake with fortification may result in excessive carbohydrate intake. During week 3, the intake of all nutrients was positively correlated with weight z-score change. Higher protein, carbohydrate, and calorie intakes were associated with better head circumference growth. None of the nutrients had any positive correlation with length growth in the whole cohort. We cannot explain the strong positive correlation between fat intake and length growth in babies with BW < 1000 g and the completely opposite result in the 1000–1249 g BW group during the second week of life. It might be related to the small sample size. A positive correlation between nutrient intake, especially protein, and better weight and head circumference growth is consistent with the results of other studies analyzing nutrient and calorie intake [36,37], although the study conducted by Falciglia et al. [38] found that higher intake of all nutrients resulted in the better growth of all parameters. It is important to note the different effects of enteral and parenteral protein intake on head circumference growth. Studies have found a positive correlation between enteral protein intake and weight gain, head circumference, and brain volumes [37,39]. A secondary analysis of the multicenter ProVide study cohort showed a negative correlation between parenteral protein intake and head circumference growth in ELBW newborns in weeks 3 and 4 [37]. Compared to the ProVide study, the decline in head circumference z-score in our ELBW group was smaller, suggesting the beneficial effect of early achievement of enteral nutrient intake close to the recommended values.

### 4.3. Growth

The EUGR in our study was 30.3%, which is lower than reports in other nutritional studies. In the literature, the EUGR has ranged from approximately 50% in large cohorts [40,41] to 66–68% in smaller studies [42,43]. A study conducted by Rigo et al. [44] showed EUGR in 21% of infants after optimizing nutrient intake. However, in this study, parenteral nutrition support was longer. Studies in which full enteral feeding was reached at a similar age to ours have reported different outcomes. The proportion of infants with EUGR was 42% in the study conducted by Rimon et al. [45] and 18% in the study conducted by Maas [32]. However, these two authors used different growth references. The high variability in the methodology of nutritional and growth studies create difficulties in defining the optimal postnatal growth of very preterm infants. In a recent publication by Fenton [46], the term of EUGR was named as a “misnomer” since it is not predictive for adverse neurodevelopmental outcomes. Moreover, very preterm infants were born with lower z-scores than the average fetus at the same gestational age [47]. Postnatal weight loss occurs soon after birth. Thus, even after regaining the birth weight and growing parallel to the curve afterward, very preterm infants will remain below their birth weight centile, and some of them might be classified as growth-restricted. Hence, it is suggested that growth monitoring immediately after birth should be used to follow growth trajectory [48]. A different approach in identifying malnutrition in neonatal population is also suggested which combines the decline in weight and length z-scores, weight gain velocity, days to regain birth weight, and nutritional intake [49]. The indicator for weight z-score change is based on a multicenter study conducted by Rochow [50] which included premature newborns with no morbidities and minimal respiratory support. These infants lost weight approximately −0.8 z-score and then maintained the growth trajectory close to that curve. The same findings were recently reported by Greenbury et al. [47] in a large cohort of very preterm infants. After initial weight loss, the weight gain trajectory stabilized along parallel percentile lines below the birth percentile for those without major morbidities. Our cohort showed a similar growth trajectory without the exclusion of infants with morbidities. A decline in the weight z-scores from birth to discharge was close to the physiological change in the weight z-score during the first week of life. The z-score for length declined throughout the hospital stay. However, at discharge, it was higher than those found in other studies [37,51]. 

Our study has strengths and limitations. The precise calculation of nutrient intakes with the analysis of human milk is the main strength of this study. On the other hand, this is a limitation causing the first week data gap, but we considered it unethical to use milk for analysis instead of feeding it to vulnerable infants in their first days of life. Another limitation of our study is analyzing OMM twice a week rather than daily, which made the nutritional calculations less accurate.

Our relatively small and heterogenous cohort also presents some limitations to our study. After dividing the cohort into three groups, the number of infants in each group was relatively small, which may have had an impact on nutrient and growth correlations. From a clinician’s point of view, it is important to foresee certain feeding interventions early in the infant’s life, since nutritional requirements and growth patterns may differ in very preterm infants but may be appropriate for gestational age infants from those who are SGA and slightly preterm.

## 5. Conclusions

Early progressive enteral feeding is well tolerated in VLBW and even in ELBW infants. Standard feeding protocol, including own mother’s colostrum and unprocessed milk, may improve in-hospital growth. The recommended enteral nutrient intake may be reached at the beginning of the second week of life. The decline of protein intake with standard fortification may compromise growth in some VLBW infants after 4 weeks of life. Thus, the safety, efficacy, and timing of individual fortification should be further investigated.

## Figures and Tables

**Figure 1 nutrients-14-01181-f001:**
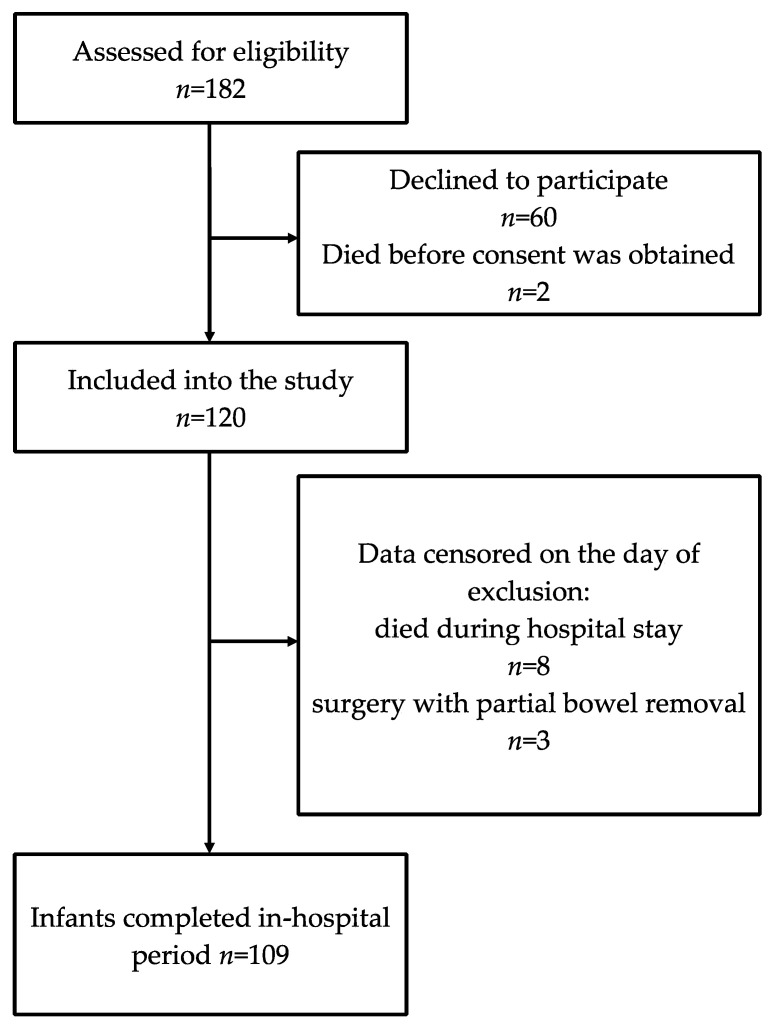
Flowchart of the study subjects’ inclusion.

**Figure 2 nutrients-14-01181-f002:**
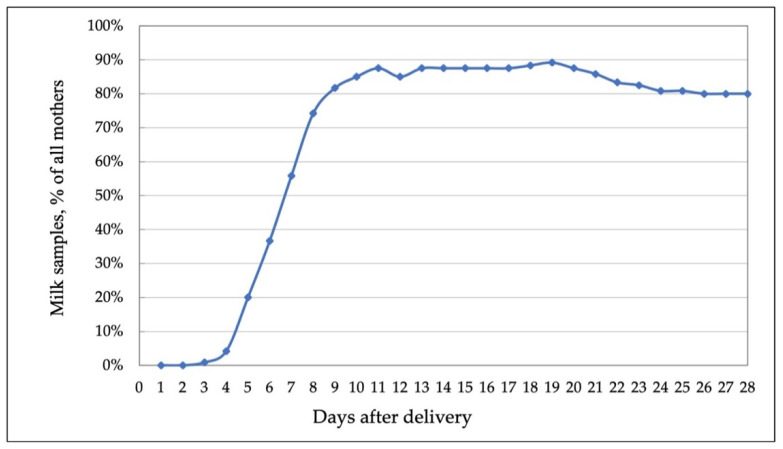
Amount of mother milk samples (% of all mothers) collected each day post-partum.

**Figure 3 nutrients-14-01181-f003:**
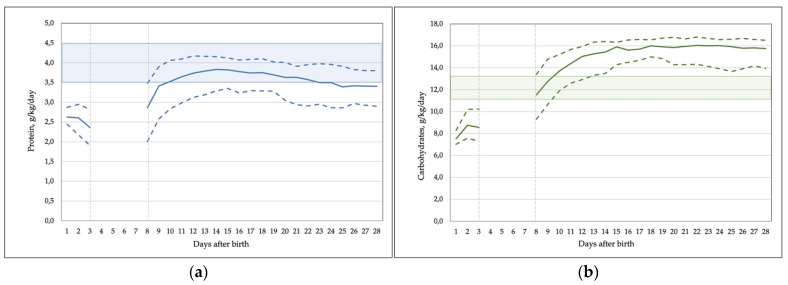
Macronutrient, energy, and fluid intake during the first 28 days. (**a**) Total daily protein intake, g/kg/day; (**b**) total daily carbohydrates intake, g/kg/day; (**c**) total daily fat intake, g/kg/day; (**d**) total daily energy intake, kcal/kg/day; (**e**) daily parenteral and enteral fluid intake, mL/kg/day. Data are median (solid line) and interquartile range (dashed lines). Shaded areas represent the range of recommended enteral nutrient intakes.

**Figure 4 nutrients-14-01181-f004:**
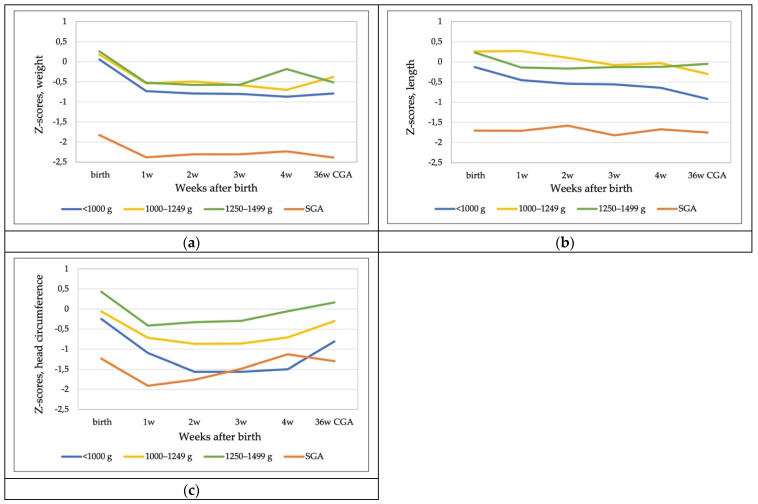
Z-scores for weight, length, and head circumference of infants of different birth weight groups and SGA infants (Fenton reference) from birth to 28 days and 36 weeks CGA or discharge. (**a**) Weight z-scores; (**b**) length z-scores; (**c**) head circumference z-scores. Data are median.

**Figure 5 nutrients-14-01181-f005:**
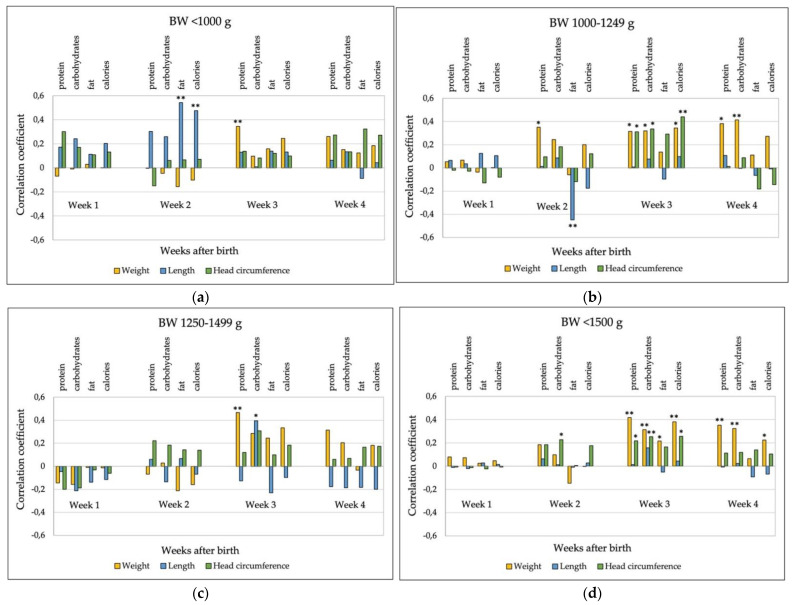
Correlations (Spearman’s rho) between nutrient and energy intake and growth during the first 4 weeks. (**a**) Birthweight group < 1000 g; (**b**) birthweight group 1000–1249 g; (**c**) birth weight group 1250–1499 g; (**d**) the whole cohort, (**e**) SGA infants. * Correlation is significant at 0.05 level; ** correlation is significant at 0.01 level.

**Table 1 nutrients-14-01181-t001:** Sample characteristics (*n* = 120).

Characteristic	Group 1 (<1000 g) *n* = 43	Group 2 (1000–1249 g) *n* = 43	Group 3 (1250–1499 g) *n* = 34
**At birth**			
Birth weight, g	830 (684–910)	1148 (1084–1204)	1370 (1311.5–1435)
Gestation, weeks	26 (25–27)	28 (27–29)	29 (28–31)
Apgar score: 1 min.	6 (3–8)	7 (5–8)	8 (7–8)
5 min.	8 (6–8)	8 (7–9)	(8–9)
Male	18 (41.9)	25 (58.1)	14 (41.2)
SGA: National reference [20]	14 (32.6)	10 (23.3)	2 (5.9)
Fenton reference [19]	6 (14.0)	4 (9.3)	2 (5.9)
Cesarean section	20 (46.5)	21 (48.8)	21 (61.8)
Multiple pregnancy	9 (20.9)	10 (23.3)	10 (29.4)
Antenatal steroids, full course	39 (90.7)	33 (76.7)	29 (85.3)
Weight Z-score [19]	−0.29 (−1.02–0.46)	0.05 (−0.7–0.7)	0.21 (−0.52–0.81)
**From birth to discharge**			
Mechanical ventilation, any	23 (53.5)	11 (25.6)	2 (5.6)
Necrotizing enterocolitis, stage II-III/SIP	6 (14.0)	1 (2.3)	1 (2.9)
Late onset sepsis	11 (25.6)	7 (16.3)	1 (2.9)
Bronchopulmonary dysplasia (O_2_ at 36 weeks CGA)	5 (11.6)	3 (7.0)	0 (0)
ROP, total	17 (39.5)	1 (2.3)	0 (0)
laser treatment	5 (20.9)	0 (0)	0 (0)
Intraventricular hemorrhage, grade ≥ III	9 (9.2)	1 (2.3)	1 (2.9)
CGA at discharge, weeks	36 (34–36)	35 (35–36)	36 (34–36)

CGA—corrected gestational age; SGA—small for gestational age; SIP—spontaneous intestinal perforation, ROP—retinopathy of prematurity. Values are median (IQR) and *n* (%).

**Table 2 nutrients-14-01181-t002:** Feeding characteristics of infants in different birth weight groups.

Characteristics	Group 1	Group 2	Group 3	*p*-Value *
First feeding, hours	6 (4–8)	3 (3–5)	4 (3–5)	<0.001 ^a,b^
First colostrum, hours	9 (5–21)	5 (3–15)	8 (5–21)	0.091
Duration of PN, days	7 (6–9)	6 (5–7)	5 (4–6)	<0.05 ^a,b,c^
Donor human milk, days	2 (1–3)	3 (2–4)	4 (3–12)	<0.05 ^b,c^
Fortification started, days	9 (7–10)	8 (7–9)	7 (6–8)	<0.01 ^b,c^
Days to regain birth weight	10 (8–14)	11 (9–12)	12 (9–14)	0.359

PN—parenteral nutrition. Data are median (interquartile range); * ^a^—Group 1 vs. Group 2, ^b^—Group 1 vs. Group 3, ^c^—Group 2 vs. Group 3.

**Table 3 nutrients-14-01181-t003:** Change in z-scores of weight, length, and head circumference (HC) in different birth weight groups from birth to 28 days and 36 weeks CGA or discharge.

Time Interval	Parameter	Birth Weight Group			*p*-Value *
		Group 1	Group 2	Group 3	
Week 1	Weight	−0.65 (−0.84–−0.36)	−0.66 (−0.84–−0.52)	−0.74 (−1.01–−0.59)	0.06
	Length	−0.28 (−0.58–0.02)	0.07 (−0.49–0.35)	−0.26 (−0.49–0.03)	0.074
	HC	−0.75 (−1.32–−0.56)	−0.67 (−1.09–−0.39)	−0.83 (−1.20–−0.40)	0.194
Week 2	Weight	−0.01 (−0.09–0.04)	−0.02 (−0.09–0.09)	−0.02 (−0.10–0.11)	0.749
	Length	−0.02 (−0.19–0.18)	−0.15 (−0.44–0.12)	−0.05 (−0.38–0.14)	0.239
	HC	−0.25 (−0.46–−0.06)	0.02 (−0.28–0.12)	0.09 (−0.14–0.23)	<0.05 ^a,b^
Week 3	Weight	−0.07 (−0.16–0.03)	−0.05 (−0.17–0.07)	0.05 (−0.10–0.10)	0.077
	Length	−0.19 (−0.35–0.16)	−0.01(−0.26–0.19)	−0.11 (−0.32–0.26)	0.476
	HC	0.00 (−0.20–0.16)	0.01 (−0.14–0.18)	0.08 (−0.00–0.28)	0.371
Week 4	Weight	−0.06 (−0.10–0.05)	−0.04 (−0.16–0.13)	0.12 (0.00–0.20)	<0.01 ^b,c^
	Length	−0.16 (−0.40–0.03)	0.04 (−0.24–0.28)	0.03 (−0.20–0.17)	0.056
	HC	0.08 (−0.22–0.15)	0.17 (−0.02–0.32)	0.25 (0.00–0.38)	0.013 ^b^
birth to 36 weeks CGA	Weight	−0.64 (−1.41–−0.25)	−0.62 (−1.04–−0.26)	−0.82 (−1.01–−0.33)	0.812
	Length	−0.61 (−1.17–−0.25)	−0.47 (−0.83–0.04)	−0.36 (−0.74–−0.02)	0.092
	HC	−0.41 (−1.02–−0.02)	−0.04 (−0.78–0.34)	−0.27 (−0.79–0.20)	0.171

CGA—corrected gestational age. * ^a^—Group 1 vs. Group 2, ^b^—Group 1 vs. Group 3, ^c^—Group 2 vs. Group 3.

## Data Availability

The data generated and analyzed during the current study are not publicly available but are available from the corresponding author on reasonable request.

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
