# Peer review of "Nutrient Intake with Early Progressive Enteral Feeding and Growth of Very Low-Birth-Weight Newborns"

_nutrients, 2022, doi:10.3390/nu14061181_

Round 1

Reviewer 1 Report

  1. Lines 26 and 259 and elsewhere: recommend replacing “own mother’s milk” with “human milk” to encompass both maternal and donor milk
  2. Table 1: provide birth weight Z-scores
  3. Table 1: provide break-down for the three birthweight cohorts
  4. Line 188: further quantify the amount of donor milk all of the infants received and compare the nutrient analyses of donor milk and own mother’s milk
  5. If readily available, consider adding historical data regarding discharge weights or Z-scores or percentage of infants with EUGR to add context to any changes seen after institution of the current progressive nutrition protocol.
  6. Grammar is notably worse throughout the discussion section
  7. Lines 267-268: should be moved to the results section

Reviewer 2 Report

The authors performed a prospective, non-interventional observational study aimed to assess nutrient intake during the first four weeks of life in the frame of an early progressive enteral feeding and to evaluate its impact on in-hospital growth of VLBW preterm infants.

There are several issues to be addressed. Below are my specific comments.

MAJOR ISSUES

  1. As authors recognized in the discussion session, nutritional requirements and growth patterns may differ in very preterm but AGA infants from those slightly preterm but IUGR. Please provide separate growth trajectory details, nutrient/fluid/energy intake’s trend during first 28 days and correlation analysis for IUGR infants in your cohort.
  2. Authors support the conceptual framework according to which ideally, postnatal growth of premature infants should be equivalent to intrauterine growth of the normal human fetus. However, as other authors recently pointed out, it may not be completely valid, as preterm birth is an abnormal phenomenon, and the optimal pattern of postnatal growth- the pattern associated with healthy long-term outcomes- is unknown. Moreover, attempting to mimic the pattern of intrauterine growth may produce an accelerated weight gain. Authors should discuss these two different conceptual frameworks and place their finding in the context of both of them. Suggested reference that should be taken into account are:

Greenbury SF, Angelini ED, Ougham K, Battersby C, Gale C, Uthaya S, Modi N. Birthweight and patterns of postnatal weight gain in very and extremely preterm babies in England and Wales, 2008-19: a cohort study. Lancet Child Adolesc Health. 2021 Oct;5(10):719-728. doi: 10.1016/S2352-4642(21)00232-7. Epub 2021 Aug 25. PMID: 34450109.

Villar J, Giuliani F, Barros F, Roggero P, Coronado Zarco IA, Rego MAS, Ochieng R, Gianni ML, Rao S, Lambert A, Ryumina I, Britto C, Chawla D, Cheikh Ismail L, Ali SR, Hirst J, Teji JS, Abawi K, Asibey J, Agyeman-Duah J, McCormick K, Bertino E, Papageorghiou AT, Figueras-Aloy J, Bhutta Z, Kennedy S. Monitoring the Postnatal Growth of Preterm Infants: A Paradigm Change. Pediatrics. 2018 Feb;141(2):e20172467. doi: 10.1542/peds.2017-2467. Epub 2018 Jan 4. PMID: 29301912.

  1. Authors declared that enteral feeding was well tolerated (L26-L372- L261). However, they did not clearly state the definition of feeding tolerance used in their unit and the protocol to manage it. Please provide these details.
  2. Why authors exclude infants discharged before 28 days? Authors did not specify a 28-day length of stay in their experimental design.
  3. Several definitions of EUGR have been proposed. How did authors choose the one described at line 163? Please detail the choice.
  4. Regarding OMM nutritional analysis, OMM was analysed twice a week and for the days between measurements values were calculated with exponential method between two nearest values. Please clearly address this point among study’s limitations.
  5. L19 “Actual nutrient intakes were assessed daily for the first 28 days”. Please rephrase the sentence because of the OMM sampling gap between day 4 to 7 and because of OMM nutritional analysis method.

MINOR ISSUES

  1. Why authors referred exclusively to gavage feeding? Would it be more appropriate “enteral” feeding which include different possible enteral feeding techniques?
  2. L68 “Impact to”, please correct into “impact on”
  3. L194, 195. How many infants interrupted enteral feeding before reaching full enteral feeds? And why two infants needed EN to be interrupted after reaching FEF? Please describe it in the manuscript.
  4. Figure 3 Please plot for each panel the recommended values considered.

Round 2

Reviewer 2 Report

Authors satisfactorily addressed my comments. The introduction of growth trajectories of SGA infants and visual representation of recommended enteral nutrient intake values  made the overall picture more detailed and added interesting insights.